# Meningococcal Disease and Related Vaccinations: Knowledge, Attitudes, and Practices among Healthcare Workers Who Provide Care to Patients with Underlying High-Risk Medical Conditions

**DOI:** 10.3390/vaccines8030543

**Published:** 2020-09-18

**Authors:** Gabriella Di Giuseppe, Concetta P. Pelullo, Giorgia Della Polla, Maria Pavia

**Affiliations:** Department of Experimental Medicine, University of Campania “Luigi Vanvitelli”, 80138 Naples, Italy; gabriella.digiuseppe@unicampania.it (G.D.G.); concettapaola.pelullo@unicampania.it (C.P.P.); giorgia.dellapolla@unicampania.it (G.D.P.)

**Keywords:** healthcare workers, high-risk medical conditions, Italy, knowledge, meningococcal disease, meningococcal vaccinations, survey

## Abstract

This cross-sectional study assessed knowledge, attitudes, and practices regarding meningococcal disease and related vaccinations among healthcare workers (HCWs) who provided care to patients with underlying high-risk medical conditions. A total of 411 HCWs returned the survey. Only 35% of the respondents had a good knowledge about the incidence and lethality of meningococcal disease, the most frequent serogroups in Italy and the diseases or conditions that expose patients to a high-risk of severe complications caused by meningococcal disease. Vaccination against meningococcal disease was perceived to be highly effective by 38.4% of participants, very safe by 36.2%, and 82% agreed or strongly agreed that HCWs should promote adherence to recommended vaccinations even in hesitant patients. Moreover, 34.1% recommended meningococcal vaccinations to all eligible patients and the results of the multivariate analysis showed that older HCWs, who work in pediatric/neonatal wards, have good knowledge about meningococcal vaccinations, have a favourable attitude towards vaccinations, and do not need additional information about meningococcal vaccinations, were more likely to recommend meningococcal vaccinations to all eligible patients. Interventions aimed at the enhancement of knowledge and awareness of HCWs who provide care to these patients on the benefits of meningococcal vaccinations are warranted.

## 1. Introduction

Meningococcal disease, although rare, is recognized as a severe worldwide public health issue [1]. In Europe, 3221 cases of invasive meningococcal disease and 282 deaths were confirmed in 2017, with an incidence rate of 0.6/100.000 [2]. In Italy, in 2018, there were 170 reported cases and there was a reduction in the incidence rate from 0.33 cases/100.000 inhabitants in 2017 to 0.28 cases/100.000 inhabitants in 2018 [3].

Most of the burden of the disease, caused by the bacterium *Neisseria meningitidis*, is attributable to serogroups A, B, C, W135, X and Y, and although mainly affecting early childhood, it often occurs also in children and young adults who live in overcrowded conditions [4].

The most effective strategy to prevent meningococcal disease is through vaccination, which is currently available against serogroups A, B, C, Y and W135 [5]. In children, the vaccination schedule suggests quadrivalent ACWY or monovalent C anti-meningococcal vaccination after the first year of life and meningococcal B vaccine from the third month of life [5], and in adolescents, a booster with quadrivalent vaccine or a first shot for those who have never been vaccinated in childhood [5], and more recently, meningococcal B vaccine for 12–18 year olds [6]. In addition to children and adolescents, several target population groups have been recommended anti-meningococcal vaccinations, among which subjects who attend places at high-risk of contagion (nursery schools, college, etc.), travelers to meningococcal disease endemic areas, health care workers (HCWs) in particular settings (who work in microbiology laboratories, emergency and intensive care services, and infectious disease departments), and patients with specific underlying medical conditions (thalassemia, immunosuppression, diabetes, severe chronic liver disease, congenital or acquired immuno-deficiencies, etc.) [5,6].

It is crucial for these patients to be vaccinated because their immune system is weaker and they are more likely to develop complications in their condition, which may involve long-term illness, hospitalization, and even death, when they acquire certain vaccine-preventable diseases, such as meningococcal disease [7]. Nevertheless, the vaccination coverage of these patients is suboptimal [8,9,10]. HCWs who provide care to these patients for their underlying medical conditions are key elements in meningococcal vaccination strategies targeted at improving adherence in this high-risk population, since it is well known that they are one of the most influential and trusted source of advice on health-related issues [8,11,12,13,14,15,16], but these professionals may lack knowledge or perceive barriers to their role in the prevention of meningococcal disease through vaccination in these groups of patients.

Moreover, most studies evaluating knowledge, attitudes and behaviors about meningococcal disease and related vaccinations have been conducted in the general population [17,18,19], or general practitioners (GPs) and paediatricians [17,20,21], whereas, to the best of our knowledge, there are no investigations that have been focused on HCWs who provide care to these patients for their underlying medical conditions.

Thus, the primary objective of this study was to explore the knowledge, attitudes, and practices about meningococcal disease and related vaccinations among HCWs who provide care to patients with underlying high-risk medical conditions in Italy.

## 2. Materials and Methods

### 2.1. Study Design

The cross-sectional survey lasted from June until July 2020, and a total of 4 public hospitals were randomly selected across all hospitals in Campania Region, Italy. Before the start of the data collection, the Directors of the selected hospitals were sent a letter requesting collaboration in the survey, once the approval to conduct the study was obtained, HCWs who were employed in the units of pediatrics, oncology, hematology, endocrinology, nephrology, internal medicine and neonatology were informed of the survey and were invited to participate. The research team distributed the questionnaire to all HCWs, including a cover letter explaining the purpose of the study, and participation was voluntary and full confidentiality was assured. Anonymization of any personal identifier or situation was granted and a written informed consent form was also included to process personal data. Follow-up visits were made to each hospital to ensure maximum recruitment of the sampled HCWs. There were no incentives offered to those who participated in the survey. A sample size of 361 HCWs was calculated by using single population proportion formula with the assumption that 17% of HCWs [17,20,21,22] would recommend meningococcal vaccinations, a confidence level of 95%, a margin of error at 5%, and a response rate of 60%. The questionnaire, developed by the research team, and based on previously validated questionnaires [19,23], was pilot tested prior to the beginning of the survey in order to ensure that the questions were understood as intended and to omit or reformulate the questions that were misinterpreted. A convenience sample of 50 HCWs was used and the data from the pilot study were not included in the final analysis. Ethical approval was obtained from the Ethics Committee of the Teaching Hospital of the University of Campania ‘‘Luigi Vanvitelli” (N° 0008660/i 15/04/2020).

### 2.2. Survey Instrument

The structured questionnaire was self-administered and consenting participants were asked to answer questions on the following five themes: (1) socio-demographic and professional characteristics (gender, age, marital status, number of children, occupation, ward, years in practice); (2) knowledge regarding meningococcal disease and related vaccinations; (3) attitudes towards the effectiveness, usefulness and safety of meningococcal vaccines, as well as towards the role of HCWs in the promotion of anti-meningococcal vaccinations to eligible patients; (4) behaviors concerning the recommendation of meningococcal vaccines to patients at risk for severe complications of their underlying medical condition in the presence of meningococcal disease (thalassemia, sickle cell anemia, immune disorders, type 1 diabetes, chronic kidney disease, congenital immunodeficiencies, Human Immunodeficiency Virus (HIV) infection, chronic liver disease, Human Papillomavirus (HPV) infection and asplenia) [5], and about adherence to recommended vaccinations for HCWs; and (5) sources of information about meningococcal vaccines and the need for additional information. The section about knowledge included four close-ended questions about meningococcal disease incidence, lethality, endemic serotypes in Italy and risk groups for severe complications, to whom meningococcal vaccine is recommended. The section on attitudes explored beliefs about the role of HCWs in encouraging patients to undergo recommended vaccinations, and in particular meningococcal vaccinations, on a 5-point Likert scale with options from “strongly disagree” to “strongly agree”, perceptions about the safety and the effectiveness of meningococcal vaccines, as well as the usefulness of an HCWs vaccination to avoid the transmission of vaccine preventable diseases to their patients on a 10-point Likert scale with options from “not safe/effective/useful” to “completely safe/effective/useful”. The section investigating HCWs’ practices consisted of three parts: the first part investigated HCWs’ collection of information about their patients’ immunization status; the second explored HCWs’ recommendation of meningococcal vaccine to patients with thalassemia, sickle cell anemia, immune disorders, type 1 diabetes, chronic kidney disease, congenital immunodeficiencies, HIV infection, chronic liver disease, HPV infection, and asplenia; and the adherence of HCWs to recommended vaccines was investigated in the third part of the section. The response options were in a ‘‘yes” or ‘‘no” format. The last section included questions about the sources and need for information regarding meningococcal vaccines.

### 2.3. Statistical Analysis

All statistical analyses were performed with the Stata software, version 15 [24]. Firstly, a descriptive analysis was used to explore the main characteristics of the sample. Secondly, chi-square and Student’s *t*-tests were conducted to assess the association between each independent variable and the different outcomes of interest. Then, multivariate stepwise logistic models with backward elimination were constructed to identify factors associated with the following outcomes of interest: good knowledge about meningococcal disease and related vaccinations (no = 0; yes = 1) (Model 1); belief that HCWs should promote adherence to recommended vaccination even in hesitant patients (no = 0; yes = 1) (Model 2); and recommending the meningococcal vaccinations to all eligible patients with underlying high risk medical conditions (no = 0; yes = 1) (Model 3). To assess the level of knowledge, an overall knowledge score was constructed by assigning 1 point for each correct answer to the questions in the knowledge section of the questionnaire. The total knowledge score ranged from 0 to 18. Then, the overall median knowledge score was calculated and a ≤50th percentile score was considered poor knowledge, whereas a >50th percentile score was good knowledge. The outcome investigating favorable attitudes was constructed by separating those who strongly agreed with the statement “HCWs should promote adherence to recommended vaccinations even in hesitant patients” versus all others. The variables that showed a *p* < 0.25 at the univariate analysis, as well as those that were considered potential determinants of the outcomes of interest were chosen to be included in the models by the stepwise procedure with backward elimination [25]. The following independent variables were included in all models: gender (male = 0; female = 1), age, categorical, in years (<36 = 1; 36–45 = 2; 46–55 = 3; >55 = 4), healthcare profession (nurse = 0; physician = 1), working in pediatric/neonatal wards (no = 0; yes = 1), scientific journals, educational activities or professional associations as sources of information about the meningococcal vaccines (no = 0; yes = 1), and need of additional information about meningococcal vaccines (no = 0; yes = 1). In Models 1 and 2, the variable length of practice in the present unit in years (continuous) was also included. In Models 2 and 3, the variable good knowledge about meningococcal disease and related vaccinations was also included. In Model 3, the variable belief that HCWs should promote adherence to recommended vaccinations in hesitant patients, was also included. The significance levels for the inclusion and elimination of the variables by the stepwise procedure were set at *p* = 0.2 and *p* = 0.4 [25]. Odds ratios (ORs) and 95% confidence intervals (CIs) were calculated. No imputation of missing data was performed. All statistical tests were two-tailed, and the level of statistical significance was set at *p* ≤ 0.05.

## 3. Results

### 3.1. Study Population

Of the 740 HCWs invited to participate in the study, 411 agreed and returned the survey for an overall response rate of 55.5%. The mean age was 47.4 years (range 25–70), 57.2% were females, the majority (70.6%) were nurses, 76.6% worked in medical wards, and the mean length of practice in the present unit was 10.9 (0–40) years (Table 1). No substantial differences of HCWs’ socio-demographic and professional characteristics were found across the selected hospitals.

### 3.2. Knowledge and Attitudes on Meningococcal Disease and Related Vaccinations

The majority (74.8%) were aware that the incidence of meningococcal disease is low in Italy, only 39.9% knew that the lethality rate ranges from 5% to 10%, whereas the remaining overestimated the severity of prognosis. Moreover, when asked to indicate the most frequent serogroups responsible for meningococcal disease in Italy, 60.8% and 57.9% correctly identified B and C, 49.9% both, while 26% reported they did not know. Furthermore, when asked about the diseases or conditions that expose patients to a high-risk of severe complications caused by meningococcal disease, respondents reported, in descending order, immune disorders (79.8%), congenital immunodeficiencies (52.1%), HIV infection (45%), asplenia (38.9%), type 1 diabetes (21.9%), chronic liver disease (20.4%), thalassemia (18.2%), sickle cell anemia (16.3%), and chronic kidney disease (15.1%). Overall, only 35% of the respondents had a good knowledge about the incidence and lethality rate of meningococcal disease, which are the most frequent serogroups in Italy and the diseases or conditions that expose patients to a high-risk of severe complications caused by meningococcal disease. Multiple logistic regression analyses showed that physicians (OR = 2.1; 95% CI = 1.24–3.54), having fewer years of practice in the present unit (OR = 0.96; 95% CI = 0.93–0.99), and having received information from scientific journals, scientific activities or professional associations (OR = 2.03; 95% CI = 1.23–3.35), and who reported need of additional information about meningococcal vaccinations (OR = 2.8; 95% CI = 1.37–5.75), were significantly more likely to have good knowledge about meningococcal disease and related vaccinations, whereas younger HCWs (<36 years) (OR = 0.38; 95% CI = 0.15–0.97) had a significantly lower knowledge than those >55 years old (Model 1 in Table 2).

A favorable attitude towards vaccinations was shown by 82% of the participants who agreed or strongly agreed that HCWs should promote adherence to recommended vaccinations even in hesitant patients. Vaccination against meningococcal disease was perceived to be highly effective by 38.4%, very safe by 36.2% of participants, and 44.5% believed that these vaccinations are very useful for HCWs to protect their high-risk patients. Furthermore, respondents reported that they agree or strongly agree that HCWs should promote meningococcal vaccinations in patients with immune disorders (81.5%), HIV infection (74.9%), congenital immunodeficiencies (73.7%), asplenia (65.9%), type 1 diabetes (65.4%), chronic liver disease (62.1%), chronic kidney disease (59.8%), sickle cell anemia (56.9%), and thalassemia (56.2%). Results of the multiple logistic regression showed that respondents who believed that HCWs should promote adherence to recommended vaccinations even in hesitant patients were significantly more likely to work in pediatric/neonatal wards (OR = 4.81; 95% CI = 2.47–9.37), to have fewer years of practice in the present unit (OR = 0.96; 95% CI = 0.93–0.99), to have been immunized against meningococcal disease (OR = 3.03; 95% CI = 1.65–5.53), and to have no need of additional information about meningococcal vaccinations (OR = 0.4; 95% CI = 0.21–0.79). Conversely, respondents aged 36–45 years (OR = 0.4; 95% CI = 0.2–0.8) and 46–55 years (OR = 0.44; 95% CI = 0.24–0.81) were less likely to have this favourable attitude than those in the age group over 55 years old (Model 2 in Table 2).

### 3.3. Behaviors Related to Meningococcal Disease and Related Vaccinations

Only 26% of respondents reported that during their work activity they verified the need of vaccinations of their patients, and an even lower proportion (12.7%) considered very high or high the acceptability of meningococcal vaccinations by their patients. Furthermore, HCWs declared they recommended, during their work activity, meningococcal vaccines to patients with immune disorders (63.5%), HIV infection (57.9%), congenital immunodeficiencies (55.5%), asplenia (54%), type 1 diabetes (48.7%), chronic liver disease (47.7%), chronic kidney disease (43.8%), sickle cell anemia (42.6%), and thalassemia (42.6%). A proportion ranging from 8% for HIV infection to 15.3% for sickle cell anemia, declared that the reason for not recommending meningococcal vaccinations was that they are not recommended for that specific condition. Overall, 97.1% had received at least one and 3.6% all the vaccinations recommended for HCWs. In particular, respondents’ self-reported adherence to recommended vaccinations was high for hepatitis B (91.5%), whereas it was lower for tetanus (67.1%), diphtheria (51.8%), pertussis (41.4%), measles (35.3%), rubella (34.5%), chickenpox (32.4%), mumps (31.6%), and meningococcal disease (21.2%). Finally, only 18.2% had received the influenza vaccination in the previous epidemic season. One third (34.1%) of respondents recommended meningococcal vaccinations to all eligible patients with underlying high-risk medical conditions, and the results of the multiple logistic regression showed that those who work in pediatric/neonatal wards (OR = 2.75; 95% CI = 1.51–4.99), have a good knowledge about meningococcal disease and related vaccinations (OR = 1.72; 95% CI = 1.02–2.88), believe that HCWs should promote adherence to recommended vaccinations even in hesitant patients (OR = 4.04; 95% CI = 2.39–6.83), and do not need additional information about meningococcal vaccinations (OR = 0.47; 95% CI = 0.26–0.85), were more likely to recommend the meningococcal vaccinations to all patients with underlying high-risk medical conditions, whereas those aged 36–45 years (OR = 0.5; 95% CI = 0.27–0.93) were less likely than those >55 years old (Model 3 in Table 2).

### 3.4. Sources of Information

The vast majority (75.4%) of the HCWs had obtained information about meningococcal vaccinations, 50.7% from internet, followed by scientific journals (37.1%), mass-media (35.5%), continuing medical education courses (33.2%), congresses (30.6%), colleagues (21%), and professional associations (13.2%). Moreover, 48.7% and 42.3% considered their knowledge about meningococcal disease and related vaccinations strategy to be very poor/unacceptable, respectively, only 29% believed that the information received about prevention of meningococcal disease was useful, and 79.3% reported the need for additional information on this topic.

## 4. Discussion

To our knowledge, this is the first study that has thoroughly investigated knowledge, attitudes, and practices about meningococcal disease and related vaccinations among HCWs who provide care to patients with underlying high-risk medical conditions for whom these vaccinations are recommended, whereas most of the studies have involved GPs [17,20] and pediatricians [21]. The results bring novel and interesting knowledge that may have implications on the strategies aimed at improving the coverage of meningococcal vaccinations in several categories of patients. Indeed, in recent decades, the inclusion of vaccines for patients with underlying high-risk medical conditions in immunization programs has provided an opportunity to protect these patients from numerous vaccine-preventable diseases, but coverage rates remain far from satisfactory. Therefore, all efforts aimed at detecting and analyzing eventual missed opportunities for recommendation or administration of meningococcal vaccinations may be useful to improve coverage rates.

### 4.1. Knowledge and Attitudes on Meningococcal Disease and Related Vaccinations

Some interesting remarks should be underscored in the results, addressing HCWs’ knowledge about meningococcal diseases and related vaccinations. Respondents tended to overestimate the lethality of meningococcal disease, and to mainly consider immune disorders as the most high-risk conditions for severe complications related to meningococcal disease, underestimating the role of other disorders, such as chronic kidney or liver disease. These findings suggest that HCWs’ knowledge is focused on all vaccine-preventable diseases and not specifically to meningococcal disease and related vaccinations. This is not surprising since a considerably lower level of knowledge was observed in a study performed in Turkey among pediatric and adult specialists, wherein asplenia/splenectomy (9.7%) and immunodeficient patients (9.7%) were identified among the risk groups recommended for anti-meningococcal vaccination [22]. Nevertheless, almost half of the participants evaluated their knowledge about meningococcal disease (48.7%) and related vaccination strategy (42.3%) to be very poor and a large majority of participants reported the need for additional information on this topic. Interesting implications apply to determinants of good knowledge. Indeed, we found that those who had received information from scientific journals or scientific activities or professional associations and had fewer years of practice revealed a higher level of knowledge, suggesting that the quality of information as well as a more recent exposure to undergraduate education testified by fewer years in practice have influence on knowledge. These results are comfortable and support the opportunity to reinforce undergraduate and continuing education on this topic. The finding that physicians expressed a higher level of knowledge compared to nurses is of concern, since nurses are frontline caregivers, and there is evidence that they are one of the most trusted source of information to help patients make informed decisions about vaccinations [26,27]. However, this result is not surprising since the curricula of the degrees of nursing and medicine are quite different in this area, and this highlights the need to improve pre and post-graduate nursing courses in regards to these topics. These findings are also aligned with previous studies that have underlined the opportunity to evaluate nursing education curricula to enhance the focus on prevention and vaccinations [14,23,28].

An encouraging finding is the positive attitude towards vaccinations demonstrated by a large proportion of respondents (82%), who agreed or strongly agreed that HCWs should promote adherence to recommended vaccinations even in hesitant patients, whereas it was worrying that only slightly more than one third of HCWs perceived vaccinations against meningococcal disease to be very effective (38.4%) and safe (36.2%), confirming the finding of a scarce attention devoted specifically to meningococcal vaccinations. Additionally, similar to the responses on knowledge, the role of meningococcal vaccinations was mostly perceived to be effective in patients with immune disorders, whilst for other conditions, there was poor awareness of the beneficial preventive effects of these vaccinations.

In common with good knowledge, fewer years of practice were positively associated with a favorable attitude towards vaccinations, reassuring that positive attitudes are influenced by correct knowledge, whereas the finding that this positive attitude was significantly more frequent among HCWs working in pediatric/neonatal wards demonstrates that HCWs continue to have a misperception, to date, that vaccinations are mainly a prerogative of the pediatric age. As expected, and consistent with a previous study [28], those who had undergone meningococcal vaccinations had a more positive attitude towards promoting recommended vaccinations to hesitant patients, providing support to the opportunity to favor vaccinations in HCWs as an enabling factor to the recommendation of vaccinations to patients.

### 4.2. Behaviors Related to Meningococcal Disease and Related Vaccinations

One of the main findings of this study is that only one fourth (26%) of HCWs verify the patients’ need for vaccinations and only one third recommends meningococcal vaccinations to all eligible patients with underlying high-risk medical conditions. It is well established that any clinical encounter with these patients should be taken as a new opportunity to promote and/or provide vaccinations, and therefore our study highlights that there is plenty of room for improvement in order to reduce these missed opportunities for meningococcal vaccinations. This result, although concerning, is not surprising, since it has already been explored in different contexts, mainly related to missed opportunities for seasonal influenza vaccinations in patients with underlying medical conditions [29,30], urging the need for initiatives that frame vaccinations as part of the clinical management of these patients. Indeed, evidence-based strategies aimed at the improvement of adult immunization coverage assign a key role to healthcare providers that should (1) assess the vaccination status of patients at every clinical encounter, (2) recommend needed vaccines, (3) offer recommended vaccines or refer patients to vaccine providers, and (4) document the vaccines administered [31]. Within this context, the results of the multivariate analysis modelling determinants of having recommended meningococcal vaccinations while managing eligible patients have provided very useful cues to the design of interventions aimed at the enhancement of HCW’s role in this issue. Indeed, both good knowledge of meningococcal vaccinations and favourable attitudes were positively associated with the correct behaviour of having recommended meningococcal vaccinations, suggesting that all evidence-based initiatives focused on improving these predisposing factors may be effective to pursue this practice among HCWs. Several studies have demonstrated that HCWs have a key role in the decision to adhere to the vaccinations of their patients [8,16,23,32,33,34,35,36], since patients trust them and are more inclined to be compliant with medications as well as with vaccinations when they are actively promoted and recommended by their caregivers [37,38]. Based on these results, increasing knowledge and awareness of HCWs who provide care to patients with underlying high-risk medical conditions on the benefits of meningococcal vaccinations for these patients appears to be a promising intervention aimed at enhancing meningococcal vaccination coverage rates. Continuing education courses that are mandatory for hospital HCWs in Italy may represent an interesting opportunity for the inclusion of vaccines and vaccination strategies as training topics addressed specifically to nurses.

### 4.3. Limitations

There are some potential limitations in the study that are worthy of emphasis and should be considered when interpreting the results. First, the analyses were based on cross-sectional data, and therefore the nature of the associations limited us from drawing definitive causal conclusions about the observed relationships between determinants and outcomes. Second, self-reported behaviors can result in the overestimation of “desirable” responses. Indeed, respondents many have inflated compliance with recommendations of the meningococcal vaccinations. Moreover, the information about their vaccination status was also self-reported and not based on vaccination records. This might be prone to recall, declaration, or desirability biases; therefore, an over or underestimation of coverage could have occurred. However, we attempted to minimize these potential biases by ensuring complete respondent anonymity and confidentiality. Third, the collected responses might not be generalizable to other regions of the country since the sample was selected from a confined geographic area. Forth, we did not explore barriers to meningococcal vaccinations recommendations, and further research on this issue would be of interest.

## 5. Conclusions

Despite these limitations, the study has added valid and valuable new knowledge on a poorly investigated topic and the results may have significant implications for the improvement of meningococcal vaccination coverage in a notably under vaccinated population. Research on the effective interventions aimed at the enhancement of knowledge and awareness of HCWs who provide care to patients with underlying high-risk medical conditions on the benefits of meningococcal vaccinations is warranted.

## Figures and Tables

**Table 1 vaccines-08-00543-t001:** Participants’ socio-demographic and professional characteristics.

Characteristics	*n* **	%
Gender		
Female	233	57.2
Male	174	42.8
Age, years	47.4 ± 10.1 (25−70) *	
<36	60	15.5
36–45	93	24
46–55	142	36.7
>55	92	23.8
Marital status		
Married/cohabitants	262	63.7
Unmarried/separated/divorced/widowed	149	36.3
Children		
No	150	36.5
Yes	261	63.5
Profession		
Nurses	286	70.6
Physicians	119	29.4
Length of practice in the present unit, years	10.9 ± 9.9 (0−40) *	
Ward		
Pediatric/neonatal	84	20.8
Other	320	79.2

* Mean ± Standard deviation (Range). ** Number for each item may not add up to total number of study population due to missing values.

**Table 2 vaccines-08-00543-t002:** Multivariate logistic regression analyses to characterize factors associated with the outcomes of interest.

**Model 1. Good Knowledge about Meningococcal** **Disease and Related Vaccinations**	**OR**	**SE**	**95% CI**	***p***
Log likelihood = -200.17; χ^2^ = 43.98 (8 *df*); *p* < 0.0001				
Profession				
Nurses	1 *			
Physicians	2.1	0.56	1.24–3.54	0.005
Age (years)				
<36	0.38	0.18	0.15–0.97	0.043
36–45	0.57	0.23	0.26–1.24	0.159
46–55	1.41	0.47	0.73–2.72	0.309
>55 *	1 *			
Fewer years of practice in the present unit	0.96	0.01	0.93–0.99	0.015
Need of additional information about meningococcal vaccinations	2.8	1.03	1.37–5.75	0.005
Information from scientific journals, scientific activities or professional associations	2.03	0.52	1.23–3.35	0.006
Being immunized against meningococcal disease	1.71	0.48	0.98–2.97	0.058
**Model 2. Belief that HCWs Should Promote Adherence to Recommended Vaccinations even in Hesitant Patients**	**OR**	**SE**	**95% CI**	***p***
Log likelihood = −174.1; χ^2^ = 79.74 (8 *df*); *p* < 0.0001				
Working in pediatric/neonatal wards	4.81	1.63	2.47–9.37	<0.001
Fewer years of practice in the present unit	0.96	0.01	0.93–0.99	0.008
Age (years)				
<36	Backward elimination			
36–45	0.4	0.14	0.2–0.8	0.010
46–55	0.44	0.14	0.24–0.81	0.009
>55 *	1 *			
Being immunized against meningococcal disease	3.03	0.93	1.65–5.53	<0.001
No need of additional information about meningococcal vaccinations	0.4	0.14	0.21–0.79	0.008
Profession				
Nurses	1 *			
Physicians	1.66	0.49	0.93–2.96	0.086
Information from scientific journals, educational activities or professional associations	1.28	0.35	0.75–2.19	0.367
**Model 3. Having Recommended the** **Meningococcal Vaccinations to Patients** **with Underlying High-Risk Medical Conditions**	**OR**	**SE**	**95% CI**	***p***
Log likelihood = −203.55; χ^2^ = 88.25 (8 *df*); *p* < 0.0001				
Working in pediatric/neonatal wards	2.75	0.83	1.51–4.99	0.001
Age (years)				
<36	0.59	0.21	0.29–1.2	0.144
36–45	0.5	0.16	0.27–0.93	0.030
46–55	Backward elimination			
>55 *	1 *			
Good knowledge about meningococcal disease and related vaccinations	1.72	0.45	1.02–2.88	0.040
Belief that HCWs should promote adherence to recommended vaccinations even in hesitant patients	4.04	1.08	2.39–6.83	<0.001
No need of additional information about meningococcal vaccination	0.47	0.14	0.26–0.85	0.013
Profession				
Nurses	1 *			
Physicians	0.61	0.18	0.34–1.08	0.090
Gender				
Male	1 *			
Females	1.32	0.34	0.8–2.2	0.276

* Reference category. The following variables were removed from the models by the backward elimination procedure: gender and working in pediatric/neonatal wards (Model 1); gender and good knowledge about meningococcal disease and related vaccinations (Model 2); being immunized against meningococcal disease and information from scientific journals, educational activities or professional associations (Model 3).

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
