# Peer review of "Meningococcal Disease and Related Vaccinations: Knowledge, Attitudes, and Practices among Healthcare Workers Who Provide Care to Patients with Underlying High-Risk Medical Conditions"

_vaccines, 2020, doi:10.3390/vaccines8030543_

Round 1

Reviewer 1 Report

The autthors performed a cross-sectional study about knowledge and practices regarding of meningococcal vaccinations among healthcare workers (HCWs) among institution with high-risk medical conditions patients.

411 HCWs (working in four public hospitals of a specific italian region) completed the survery: only 35% of them had a good knowledge regarding the epidemiology of meningococcal disease and its possible complications among frail patients

The present work represents the first to investigate attitudes and practices about meningococcal disease and related vaccinations among hospital HCWs providing care to high-risk patients. Authors should be praised for their attempt to highligh and increase knowledge about a so important public health matter.

Author Response

Reviewer 1

No revisions requested.

We are very grateful to the reviewer for the extremely positive tone of the comments.

Reviewer 2 Report

The present study focusses on meningococcal disease. This is an important topic and the results found have interest.

However, I think that due to the depth of the work and other particularities I think that it is more suitable as a short report than as an article.

 Some things to point out:

  • I do not see the need to record some characteristics of the participants as the marital status or if they have or not children.
  • In the discussion, the authors explain that the level of knowledge in nurses is lower, but this is not surprising as the curricula of the degrees of nursing and medicine are quite different in this area. This is an important thing to try to improve in the pre and postgraduate studies.

Author Response

Reviewer 2

The present study focusses on meningococcal disease. This is an important topic and the results found have interest.

However, I think that due to the depth of the work and other particularities I think that it is more suitable as a short report than as an article.

Thanks for your comments. In response to this point, as highlighted in the manuscript, this is the first study that has thoroughly investigated the related topic (knowledge, attitudes, and practices about meningococcal disease and related vaccinations among HCWs who provide care to patients with underlying high-risk medical conditions for whom these vaccinations are recommended) and has provided a substantial amount of new information. The study, therefore, may stimulate further research and represent a starting point for researchers involved in similar projects. Moreover, since Vaccines has no restrictions on the length of manuscripts and reported types of publication do not include Short Reports, my co-Authors and I believe a more comprehensive description of all the investigated issues would probably be of interest to the readers, considering that information on this topic is scarce.

I do not see the need to record some characteristics of the participants as the marital status or if they have or not children.

Thanks for your comments. In response to this point, information on marital status and, particularly on children, have been collected since having children has been found to be associated to higher level of knowledge on vaccinations and to a more favorable attitude in a recent study conducted by some of us on vaccinations in HCWs (Reference No.23). However, since this was not the case in this study, according to your suggestion, we have now eliminated these variables from the models.

In the discussion, the authors explain that the level of knowledge in nurses is lower, but this is not surprising as the curricula of the degrees of nursing and medicine are quite different in this area. This is an important thing to try to improve in the pre and postgraduate studies.

(Lines 281-284) Thanks for your comments. In response to this point, we agree with the reviewer that differences in nursing and medicine degrees curricula are different and that there is place for improvement of the nursing education on these topics. This issue has now been underlined in the Discussion session.

Reviewer 3 Report

In this cross-sectional study, the authors investigated the knowledge, attitudes, and practices of healthcare workers providing care for patients with underlying high-risk medical conditions regarding meningococcal disease and related vaccination. 

There are several main concerns. 

Firstly, according to your statistical description results, I found that some variables were missing to different degrees. How to deal with the missing value? Did you impute it? If a imputation was made, how was it made? 

Secondly, logistic regressions were used to identify associated factors. Please explain the logics to screen factors (stepwise)?

Thirdly, the number of subjects included in each model is unclear. Why not group with age less than 36 in model 2 and no group with age between 46 and 55 in model 3? It make uncomparable to have an extremely uneven number of participants in each group. 

Forthly, the significance levels of the inclusion and elimination of variables through the stepwise process were set to P = 0.2 and P = 0.4, which were too large, and the general inclusion and exclusion criteria are set to P =0.05 and P =0.1, respectively. Please expain.

Finally, it is recommended that some percentage descriptions in the article can be changed to figures.

There are also several minor concerns, such as

The title of Table 2 should be clearly defined.  

Several typos.

Author Response

Reviewer 3

Firstly, according to your statistical description results, I found that some variables were missing to different degrees. How to deal with the missing value? Did you impute it? If a imputation was made, how was it made?

(Lines 154-155) Thanks for your comments. In response to this point, since for each variable the number of missing values was low, we did not impute them, and we performed the analysis using only actual values. We have now clarified this point in the methods section.

Secondly, logistic regressions were used to identify associated factors. Please explain the logics to screen factors (stepwise)?

(Lines 140-142) Thanks for your comments. In response to this point, variables selection and model building strategy was performed according to Hosmer and Lemeshow (Hosmer DW, Lemeshow S. Applied Logistic Regression. 2nd Edition, New York: Wiley; 2000). In particular, variables that exhibited a <0.25 p-value at the univariate analysis and those that were considered potential determinants of the selected outcomes were included in the models, that were performed using a stepwise procedure with backward elimination. The model building strategy has now been more thoroughly described in the methods section.

Thirdly, the number of subjects included in each model is unclear. Why not group with age less than 36 in model 2 and no group with age between 46 and 55 in model 3? It make uncomparable to have an extremely uneven number of participants in each group.

(Table 2) Thanks for your comments. In response to this point, as already mentioned, we used the stepwise procedure for model building strategy, and the independent variable “age” was categorized in several age intervals (<36=1; 36-45=2; 46-55=3 >55=4) and, as such, was included in all models. Therefore, there were some age categories that were excluded from the model by the backward elimination procedure, since they did not fulfill the criterion for inclusion, and this was the case for age group < 36 years in Model 2 and for age group 46-55 years in Model 3. To avoid misunderstanding, we have now specified this issue in the Table.

Forthly, the significance levels of the inclusion and elimination of variables through the stepwise process were set to P = 0.2 and P = 0.4, which were too large, and the general inclusion and exclusion criteria are set to P =0.05 and P =0.1, respectively. Please explain.

(Line 142) Thanks for your comments. In response to this point, it is well known that a crucial aspect of using stepwise logistic regression is the choice of an “alpha” level to judge the importance of variables to be entered or dropped from the model. According to Hosmer Lemeshaw (Hosmer DW, Lemeshow S. Applied Logistic Regression. 2nd Edition, New York: Wiley; 2000) the choice of P=0.05 for including variables in the model is too stringent, often excluding important variables from the model, and they highly recommend a value in the range from 0.15 to 0.20 or even higher. Moreover, since the value of P for exclusion has to be higher than that for inclusion, we set it at 0.4, opting for a conservative strategy. To justify our choice, we have now included the reference in the methods section.

Finally, it is recommended that some percentage descriptions in the article can be changed to figures.

Thanks for your comments. As suggested, we have changed some percentage descriptions in the article into figures.

Minor comments

The title of Table 2 should be clearly defined.

Thanks for your comments. As suggested, we have changed the title of Table 2 in “Multivariate logistic regression analyses to characterize factors associated with the outcomes of interest”.

Several typos.

Response 2: Thanks for your comments. As suggested, we have corrected typos.

Reviewer 4 Report

The authors conducted a questionnaire survey on knowledge of meningococcal vaccine among healthcare workers. It was identified that there was a statistical difference in that knowledge depending on age and work location. The study will provide important information for high-risk individuals in the future to promote vaccines.

Minor comments:

  1. Surveys have been conducted targeting four hospitals. Is there any difference in the survey results between the hospitals?
  2. The awareness of the usefulness of vaccines will change if it is taken up as a training theme at in-hospital infectious disease control workshops. Isn't it necessary to have in-hospital infection control workshops etc. for medical staff in Italy? I asked it because, in my country, the medical practitioners working in hospitals need to participate in infection control workshops more than twice a year under the law.
    In that case, please included the above point in the Discussion.

Author Response

Reviewer 4

Minor comments:

Surveys have been conducted targeting four hospitals. Is there any difference in the survey results between the hospitals?

(Line 163-164) In response to this point, we have compared HCWs’ socio-demographic and professional characteristics and no substantial differences were found among hospitals. Therefore, we have performed the analysis in an aggregated form. This has been clarified in the results section.

The awareness of the usefulness of vaccines will change if it is taken up as a training theme at in-hospital infectious disease control workshops. Isn't it necessary to have in-hospital infection control workshops etc. for medical staff in Italy? I asked it because, in my country, the medical practitioners working in hospitals need to participate in infection control workshops more than twice a year under the law. In that case, please included the above point in the Discussion.

(Lines 330-332) In response to this point, although in Italy continuing education courses in hospitals are mandatory for HCW, they may deal with extremely variable topics of interest. Therefore, we have now included in the Discussion the opportunity to include workshops on the importance of vaccines and vaccination strategies among the training themes of mandatory courses.

Round 2

Reviewer 2 Report

The authors have answered the issus arisen in the review. Therefore it could be published in the present form.

Reviewer 3 Report

none